# Association between pan-immune-inflammation value and clinical outcomes in critically ill patients with hyperlipidemia: An observational study

Xiao-Ying Huang, Xiang Zheng *

Department of Critical Care Medicine, Taihe Hospital (Hubei University of Medicine), Shiyan, Hubei, China

* syzx481@hbmu.edu.cn

## Abstract

### Background

Hyperlipidemia contributes to immune-inflammatory imbalance and poor outcomes in critical illness. The pan-immune-inflammation value (PIV), derived from neutrophil, monocyte, platelet, and lymphocyte counts, may capture this burden, but its prognostic value in critically ill patients with hyperlipidemia is unclear.

### Methods

We conducted a retrospective cohort study using MIMIC-IV (v3.1). Adults with a first ICU admission and ICD-coded hyperlipidemia, with complete blood counts within 24 h, were included. lnPIV (ln-transformed PIV) was analyzed as continuous and categorical (quartiles) exposures. Primary outcomes were 28-, 90-, and 180-day mortality; secondary outcomes included ICU and in-hospital mortality, acute kidney injury (AKI), and delirium. Survival, Cox regression, restricted cubic spline (RCS), and ROC analyses were applied.

### Results

Among 12,408 patients (mean age 70 years; 38.6% female), 1,309 died in hospital. Higher lnPIV was independently associated with 28-day (HR = 1.14, 95% CI: 1.10–1.18), 90-day (HR = 1.15, 95% CI: 1.11–1.18), and 180-day mortality (HR = 1.13, 95% CI: 1.10–1.16; all P < 0.001). RCS analysis identified a threshold at PIV ≈ 172, above which mortality risk increased sharply. The highest quartile had significantly greater in-hospital mortality risk (OR = 1.85, 95% CI: 1.50–2.29). Adding PIV to the SOFA score significantly improved discrimination (AUC increased from 0.55 to 0.67, P < 0.001) and provided positive net benefit across clinically relevant threshold probabilities. lnPIV was also associated with AKI and delirium.

**Data availability statement:** The data underlying the results presented in the study are available from the PhysioNet repository (MIMIC-IV version 3.1, at https://physionet.org/content/mimiciv/).

**Funding:** The author(s) received no specific funding for this work.

**Competing interests:** The authors have declared that no competing interests exist.

## Conclusion

Elevated PIV is independently associated with mortality and adverse outcomes in critically ill patients with hyperlipidemia. As a simple and readily available composite marker, PIV may complement existing risk scores for early risk stratification in this vulnerable population.

## Introduction

Hyperlipidemia has emerged as a global health issue reflecting an imbalance in lipid metabolism. With an approximately 39% prevalence rate among adults worldwide, the condition affects over half of the population in specific high-risk groups [1,2]. This disorder is a major independent risk factor for cardiovascular disease, stroke, and other critical conditions, posing growing public health challenges [3,4]. This is directly linked to endothelial dysfunction, chronic inflammation, and ultimately atherosclerosis resulting from abnormal lipid metabolism in hyperlipidemia patients [5]. In the stress state of critical illness, patients undergo significant alterations in lipid metabolism, characterized by persistently elevated triglyceride levels and reduced low-density lipoprotein and high-density lipoprotein levels [6]. These changes in lipid profiles are frequently overlooked by clinicians, yet their consequences can be severe. Not only can these alterations induce lipotoxicity, but they also amplify stress responses, disrupt cell membrane integrity, and accelerate metabolic breakdown, increasing the production of free fatty acids through their close association with inflammatory responses [7]. Free fatty acids, such as palmitic acid, are potent damage-associated molecular patterns that can activate the Toll-Like Receptor 4 signaling pathway, driving strong inflammatory reactions [8]. Based on the pathophysiological role of hypertriglyceridemia, it exerts significant impacts in specific clinical diseases. The incidence of acute pancreatitis (AP) caused by hyperlipidemia—especially hypertriglyceridemia—is increasing year by year in Europe and the United States [9]. Hypertriglyceridemia not only exacerbates AP caused by other etiologies but also significantly increases the recurrence rate, readmission rate, and incidence of sequelae in AP patients [10]. High triglyceride levels are an independent predictor of mortality in sepsis and are associated with elevated inflammatory markers during the course of the disease [11].

Although the clinical significance of hyperlipidemia in critical illness is widely acknowledged, there is still no tool that can accurately predict immune and inflammatory imbalances in this population. Against this backdrop, the pan-immune-inflammation value (PIV) has emerged as a novel biomarker, gradually gaining attention from researchers. Initially proposed in rectal cancer research, PIV serves as a comprehensive indicator for evaluating inflammation and immune status [12]. PIV integrates neutrophil, monocyte, platelet, and lymphocyte counts, each component can be rapidly measured through routine peripheral blood tests, making it highly feasible in clinical practice. Subsequent meta-analyses have demonstrated that PIV performs well in identifying prognostic risks across various cancers [13]. Moreover,

PIV is closely linked to adverse outcomes in a range of critical illnesses, such as sepsis, myocardial infarction, aortic dissection, and pulmonary embolism [14–17].

Although PIV is prognostic in several diseases, its value among critically ill patients with hyperlipidemia remains unknown. We therefore evaluated the association between admission PIV and mortality and complications in this population using the MIMIC-IV database.

## Methods

### Data source

This study was a retrospective cohort analysis, and the data were sourced from the Medical Information Mart for Intensive Care IV (MIMIC-IV, version 3.1) [18]. The MIMIC-IV database, developed and maintained by the Laboratory for Computational Physiology at the Massachusetts Institute of Technology, contains detailed clinical data on over 380,000 hospital admissions and more than 60,000 ICU admissions at Beth Israel Deaconess Medical Center in Boston, Massachusetts, from 2008 to 2022. The database includes patient demographics, diagnoses, laboratory tests, vital signs, medications, clinical interventions, and outcomes. MIMIC-IV has been widely used in critical care research and in the development of clinical prediction models. The MIMIC-IV database was approved by the Institutional Review Boards of the Massachusetts Institute of Technology (Cambridge, MA) and Beth Israel Deaconess Medical Center (Boston, MA). All patient data were de-identified in accordance with the Health Insurance Portability and Accountability Act (HIPAA) to protect patient privacy. The researcher completed the National Institutes of Health and Collaborative Institutional Training Initiative certification (ID: 68911069) and was granted permission to use the database. As this study used only de-identified, publicly available data, it was exempt from additional institutional review board approval and the requirement for informed consent. The data were accessed for research purposes on 07 April 2025.

### Study population, variable extraction, and outcomes

The inclusion criteria were as follows: (1) age ≥ 18 years; (2) first ICU admission during the index hospitalization; (3) diagnosis of hyperlipidemia identified by ICD-9 or ICD-10 codes; and (4) ICU length of stay ≥24 hours. The exclusion criteria were as follows: (1) missing outcome data (e.g., in-hospital mortality data missing); (2) missing or incomplete complete blood count parameters, or abnormal/unreasonable laboratory values; and (3) multiple ICU admissions during the same hospitalization, with only the first ICU admission analyzed. A total of 12,408 eligible patients were included (Fig 1).

PIV was defined as: (neutrophil count × platelet count × monocyte count)/ lymphocyte count, with all parameters measured in ×$10^9$/L, obtained from the first blood sample within 24 hours of ICU admission. To address skewed distribution and minimize the influence of extreme values, the natural logarithm of PIV (lnPIV) was used for statistical analyses. Patients were divided into quartiles based on lnPIV for between-group comparisons. The primary outcomes were all-cause mortality at 28, 90, and 180 days. The secondary outcomes included ICU mortality, in-hospital mortality, acute kidney injury (AKI), and the incidence of delirium. AKI was defined according to KDIGO criteria, and delirium was assessed using the Confusion Assessment Method for the Intensive Care Unit [19,20].

Variables were extracted using SQL queries, including demographics and comorbidities identified by ICD-9/10 codes (e.g., congestive heart failure, atrial fibrillation, hypertension, cerebrovascular disease, chronic lung disease, liver disease, renal disease, diabetes, and malignancies). Severity scores (SOFA, APS III, SIRS) were calculated within 24 hours of ICU admission. Vital signs, such as heart rate, respiratory rate, mean arterial pressure, temperature, and peripheral oxygen saturation, were measured alongside laboratory parameters like electrolyte levels, anion gap, renal function indicators, glucose levels, and complete blood counts. Glycated hemoglobin (HbA1c) levels were extracted from records within 3 months before or during hospitalization, using the value closest to ICU admission. Medication use during ICU stay (norepinephrine, glucocorticoids, statins) was recorded as binary variables from medication records. Lipid parameters

 

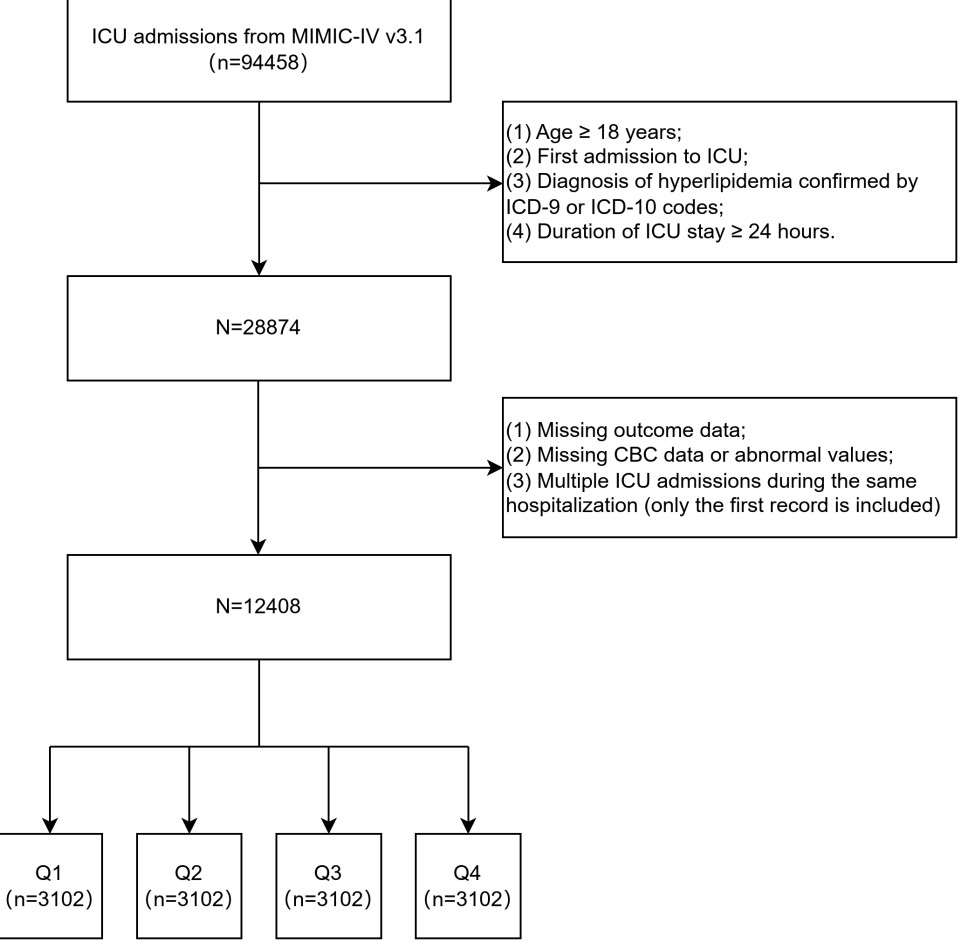

**Fig 1. A flowchart for the patient selection process.** ICU, Intensive Care Unit; ICD, International Classification of Diseases; CBC, Complete Blood Count.

(triglycerides, total cholesterol) were extracted, with hypertriglyceridemia defined as >2.3 mmol/L and hypercholesterolemia as >5.2 mmol/L based on clinical guidelines. When multiple measurements were available within 24 hours, only the initial measurement was used to ensure consistency in the analysis.

## Statistical analysis

All statistical analyses were performed using R software (version 4.2.1) and Stata 18.0 (StataCorp, Texas, USA). A two-sided P value < 0.05 was considered statistically significant. Normally distributed continuous variables were expressed as mean ± standard deviation, while non-normally distributed variables were presented as median and interquartile range. Between-group comparisons were conducted using one-way analysis of variance (ANOVA) or the Kruskal–Wallis test. Categorical variables were presented as frequency (percentage) and compared using the chi-square test or Fisher's exact test. Survival analyses were performed using the Kaplan–Meier method, with group differences assessed by the log-rank test. Multicollinearity was considered significant when the variance inflation factor was > 5. Cox proportional hazards regression models were used to evaluate the associations between lnPIV (as either a categorical or continuous variable) and the primary and secondary outcomes. Three models were constructed: (1) unadjusted; (2) partially adjusted

(demographic factors); and (3) fully adjusted. Restricted cubic spline (RCS) regression was employed to explore the non-linear relationship between lnPIV and mortality outcomes. The identified threshold was back-transformed to the original PIV scale (exp[lnPIV]) for clinical interpretability. Stratified and interaction analyses were performed to examine the consistency of associations between lnPIV and outcomes across subgroups (age < 65 or ≥65, sex, race, lipid levels, comorbidities,medication use, etc.). For covariates with missing rates <20%, multiple imputation was performed using the Multivariate Imputation by Chained Equations (MICE) algorithm. We generated 5 imputed datasets (m = 5) using classification and regression trees (method = "cart") with a random seed of 123 for reproducibility.

Density plots of observed versus imputed values for key variables demonstrated good concordance, supporting the validity of the imputation approach (S1 Fig). Variables with >20% missing values were excluded from the analysis.

## Results

### 1. Baseline Characteristics of the Subjects

A total of 12,408 critically ill patients with hyperlipidemia were included in this study, of whom 11,099 survived and 1,309 died during hospitalization (Table 1). Non-survivors were older (74.7 vs 69.8 years, P < 0.001) and had a higher proportion of males (45.5% vs 35.7%, P < 0.001). They also had higher heart rates, respiratory rates, and worse laboratory parameters (e.g., potassium, blood urea nitrogen, glucose), while survivors had higher chloride and bicarbonate levels. Non-survivors had more comorbidities (e.g., heart failure, atrial fibrillation, kidney disease, malignant tumor) and severity scores (SOFA). Analysis by lnPIV quartiles revealed that higher PIV was associated with older age, higher female proportion, and increased heart rate, respiratory rate, blood urea nitrogen, creatinine, and glucose levels (S1 Table). Comorbidities such as heart failure, lung disease, and kidney disease increased with higher lnPIV, while hypertension decreased.

### 2. Association Between lnPIV and In-hospital Mortality

Kaplan–Meier survival curves and cumulative risk plots (Fig 2) showed significant differences in survival among lnPIV quartiles at 28, 90, and 180 days (all log-rank P < 0.0001). Patients in the highest quartile (Q4) had the lowest survival, while those in the lowest quartile (Q1) consistently had the highest survival. Cumulative hazard analysis further indicated that Q4 had the highest and continuously increasing cumulative risk, followed by Q3, Q2, and Q1 with the lowest risk (all P < 0.0001, Fig 2). Cox regression analysis also confirmed that higher lnPIV was independently associated with all-cause mortality at 28, 90, and 180 days (Table 2). In the fully adjusted model (Model 3), each unit increase in lnPIV was associated with a hazard ratio (HR) of 1.26 (95% CI: 1.20–1.32) for 28-day mortality, 1.25 (95% CI: 1.20–1.32) for 90-day mortality, and 1.23 (95% CI: 1.19–1.28) for 180-day mortality (all P < 0.001). When stratified by quartiles, Q4 had the highest mortality risk compared with Q1 (HR = 1.87, 95% CI: 1.59–2.20, P < 0.001), Q3 also showed increased risk (HR = 1.29, 95% CI: 1.08–1.53, P < 0.001), while Q2 did not differ significantly from Q1 (HR = 0.97, 95% CI: 0.80–1.17, P = 0.189). The trends were consistent across all time points. Logistic regression further supported these findings. In the fully adjusted model, each unit increase in lnPIV was associated with an odds ratio (OR) of 1.26 (95% CI: 1.19–1.34, P < 0.001) for in-hospital mortality(Table 3). Compared with Q1, patients in Q4 had a significantly higher risk of in-hospital mortality (OR = 1.86, 95% CI: 1.51–2.28, P < 0.001), while the increase in Q3 was statistically significant (OR = 1.26, 95% CI: 1.02–1.57, P = 0.034).

### 3. Association Between lnPIV and Adverse Outcomes

Multivariable logistic regression showed that the risks of AKI and delirium significantly increased with higher lnPIV quartiles (Table 3). After adjusting for confounders, the risks of AKI and delirium in the third and fourth quartiles were significantly higher than those in the first quartile. The overall trend indicated progressively increasing risks with higher lnPIV (trend test P < 0.001). When analyzed as a continuous variable, lnPIV was also an independent risk factor for AKI and delirium during hospitalization.

**Table 1. Baseline characteristics by survival status.**

| Categories | Overall | Survivors | Nonsurvivors | P-value |
|---|---|---|---|---|
| | (N = 12408) | (N = 11099) | (N = 1309) | |
| Age(years) | 70.3 ± 12.0 | 69.8 ± 12.0 | 74.7 ± 11.3 | <0.001 |
| Male, n(%) | 4556(36.7) | 3961(35.7) | 595(45.5) | <0.001 |
| White, n(%) | 8445(68.1) | 7643(68.9) | 802(61.3) | <0.001 |
| Vital signs | | | | |
| Heartrate, beats/min | 82.0[74.0,95.0] | 81.0[74.0,93.0] | 92.0[79.0,108.0] | <0.001 |
| MBP, mmHg | 80.0[70.0,91.0] | 80.0[70.0,91.0] | 80.0[68.0,93.0] | 0.23 |
| Respiratory rate, beats/min | 18.0[15.0,22.0] | 17.0[15.0,21.0] | 21.0[17.0,26.0] | <0.001 |
| Laboratory tests | | | | |
| Sodium(mEq/L) | 138.0[136.0,141.0] | 138.0[136.0,141.0] | 138.0[134.0,142.0] | 0.104 |
| Potassium(mEq/L) | 4.2[3.9,4.6] | 4.2[3.90,4.60] | 4.3[3.9,4.9] | <0.001 |
| Chloride(mEq/L) | 106.0[101.0,109.0] | 106.0[102.0,109.0] | 102.0[98.0,107.0] | <0.001 |
| Bicarbonate(mEq/L) | 23.0[20.0,25.0] | 23.0[21.0,25.0] | 21.0[18.0,25.0] | <0.001 |
| Anion gap(mEq/L) | 13.0[11.0,16.0] | 13.0[10.0,15.0] | 16.0[13.0,19.0] | <0.001 |
| BUN(mg/dL) | 19.0[14.0,30.0] | 18.0[13.0,28.0] | 32.0[20.0,51.0] | <0.001 |
| Creatinine(mg/dL) | 1.0[0.8,1.4] | 1.0[0.8,1.3] | 1.40[1.0,2.4] | <0.001 |
| Glucose(mg/dL) | 127.0[107.0,160.0] | 125.0[107.0,156.0] | 150.0[115.0,200.0] | <0.001 |
| HbA1c(%) | 5.90[5.50, 6.70] | 5.90[5.50, 6.70] | 5.90[5.50, 6.60] | 0.303 |
| Comorbidities | | | | |
| CHF(%) | 4112(33.1) | 3520(31.7) | 592(45.2) | <0.001 |
| Atrial fibrillation(%) | 4688(37.8) | 4088(36.8) | 600(45.8) | <0.001 |
| Hypertension(%) | 6703(54.0) | 6138(55.3) | 565(43.2) | <0.001 |
| Cerebrovascular disease(%) | 1880(15.2) | 1612(14.5) | 268(20.5) | <0.001 |
| COPD(%) | 3097(25.0) | 2722(24.5) | 375(28.6) | 0.001 |
| Liver disease(%) | 964(7.8) | 771(6.9) | 193(14.7) | <0.001 |
| Renal disease(%) | 3065(24.7) | 2607(23.5) | 458(35.0) | <0.001 |
| Diabetes(%) | 5169(41.7) | 4606(41.5) | 563(43.0) | 0.308 |
| Malignant tumor(%) | 1402(11.3) | 1107(10.0) | 295(22.5) | <0.001 |
| Medications | | | | |
| Norepinephrine(%) | 2722(21.9) | 2003(18.0) | 719(54.9) | <0.001 |
| Glucocorticoid (%) | 2652(21.4) | 2181(19.4) | 471(41.2) | <0.001 |
| Statin(%) | 9452(76.2) | 8782(78.0) | 670(58.7) | <0.001 |
| SOFA | 2.0[0.0,3.0] | 1.0[0.0,3.0] | 2.0[0.0,4.0] | <0.001 |
| PIV | 1677.8 ± 3944.8 | 1441.6 ± 3115.3 | 3679.1 ± 7796.4 | <0.001 |
| lnPIV | 6.4 ± 1.5 | 6.3 ± 1.5 | 7.1 ± 1.8 | <0.001 |

Continuous variables are shown as mean (SD) or median (IQR), and categorical variables are shown as count (%). MBP, mean blood pressure; BUN, blood urea nitrogen; CHF, congestive heart failure; COPD, chronic obstructive pulmonary disease; SOFA, Sequential Organ Failure Assessment; PIV, pan-immune-inflammation value.

## 4. Restricted Cubic Spline (RCS) Analysis

Restricted cubic spline (RCS) analyses showed significant non-linear associations between lnPIV and mortality risk (Fig 3). For 28-day, 90-day, and 180-day mortality, the adjusted hazard ratio curves were J-shaped, remaining flat or slightly declining at lower lnPIV levels and rising progressively at higher levels (P < 0.001 for overall association; P < 0.001 for non-linearity for all timepoints).

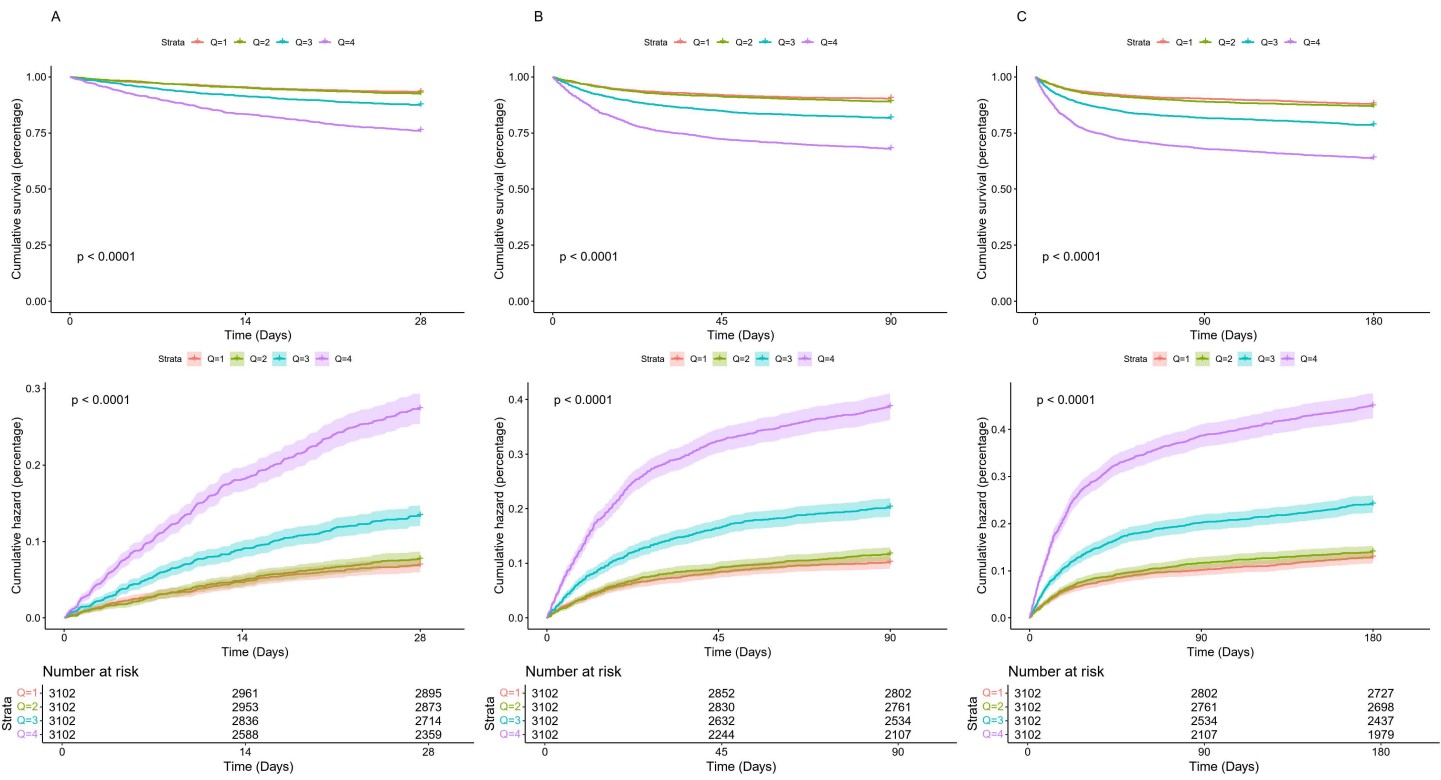

**Fig 2. The K-M survival and cumulative risk plots of (A) 28-day, (B) 90-day, (C) 180-day.**

Similarly, the risks of ICU and in-hospital mortality followed curvilinear dose–response patterns, with adjusted odds ratio curves showing little variation at lower lnPIV levels but increasing steeply as lnPIV rose (P < 0.001 for overall and for non-linearity). Segmented regression identified a critical threshold at lnPIV = 5.152, corresponding to a raw PIV value of approximately 172, for in-hospital mortality. Below this value, lnPIV was not a significant predictor (p = 0.055), whereas above it, mortality risk rose sharply (p < 0.001).This association persisted after multivariable adjustment, suggesting that PIV ≈ 172 may represent a clinically relevant cutoff for risk stratification in this population.

## 5. **ROC curve analysis**

ROC curve analysis showed that the AUC of lnPIV for predicting in-hospital mortality was 0.664 (95% CI: 0.648–0.681), higher than that of neutrophils (0.597), monocytes (0.596), platelets (0.545), and 1/lymphocytes (0.662) (Fig 4; S2 Table).
Comparisons with other composite inflammatory markers are presented in S3 Table. We further evaluated the incremental value of PIV beyond the SOFA score.Adding PIV to SOFA significantly improved discrimination, with detailed AUC comparisons, integrated discrimination improvement, and net reclassification improvement presented in S4 Table. Decision curve analysis demonstrated that the SOFA+PIV model provided higher net benefit across clinically relevant threshold probabilities compared to SOFA alone (S2 Fig).

## 6. **Subgroup analysis**

Subgroup analysis showed that the association between lnPIV and 28-day mortality was consistent across most prespecified subgroups, including age, sex, race, lipid parameters, HbA1c status, diabetes, hypertension, and cerebrovascular, renal, and liver diseases (all P for interaction > 0.05) (Fig 5).

**Table 2. Multivariable results by Cox regression analysis.**

| Categories | Model1 | | Model2 | | Model3 | |
|---|---|---|---|---|---|---|
| | HR (95% CI) | P-value | HR (95% CI) | P-value | HR (95% CI) | P-value |
| 28-day in-hospital mortality | | | | | | |
| lnPIV | 1.44(1.39, 1.49) | <0.001 | 1.40(1.36, 1.45) | <0.001 | 1.26(1.20, 1.32) | <0.001 |
| Quartile | | | | | | |
| Q1 (N = 3102) | Reference | | Reference | | Reference | |
| Q2 (N = 3102) | 1.11 (0.92, 1.34) | 0.286 | 1.09 (0.90, 1.32) | 0.366 | 0.97 (0.80, 1.17) | 0.189 |
| Q3 (N = 3102) | 1.94 (1.64, 2.29) | <0.001 | 1.82 (1.54, 2.16) | <0.001 | 1.29 (1.08, 1.53) | <0.001 |
| Q4 (N = 3102) | 3.95 (3.39, 4.61) | <0.00 | 3.58 (3.07, 4.18) | <0.001 | 1.87 (1.59, 2.20) | <0.001 |
| P for trend | | <0.001 | | <0.001 | | <0.001 |
| 90-day in-hospital mortality | | | | | | |
| lnPIV | 1.43(1.39, 1.47) | <0.001 | 1.39(1.35, 1.43) | <0.001 | 1.25(1.20, 1.32) | <0.001 |
| Quartile | | | | | | |
| Q1 (N = 3102) | Reference | | Reference | | Reference | |
| Q2 (N = 3102) | 1.14 (0.98, 1.33) | 0.096 | 1.12 (0.96, 1.31) | 0.143 | 1.11 (0.95, 1.30) | 0.175 |
| Q3 (N = 3102) | 1.98 (1.73, 2.28) | <0.001 | 1.87 (1.62, 2.15) | <0.001 | 1.40 (1.22, 1.62) | <0.001 |
| Q4 (N = 3102) | 3.81 (3.35, 4.34) | <0.001 | 3.46 (3.04, 3.94) | <0.001 | 1.93 (1.68, 2.21) | <0.001 |
| P for trend | | <0.001 | | <0.001 | | <0.001 |
| 180-day in-hospital mortality | | | | | | |
| lnPIV | 1.40(1.37, 1.44) | <0.001 | 1.37(1.33, 1.40) | <0.001 | 1.23(1.19, 1.28) | <0.001 |
| Quartile | | | | | | |
| Q1 (N = 3102) | Reference | | Reference | | Reference | |
| Q2 (N = 3102) | 1.08 (0.94, 1.25) | 0.266 | 1.07 (0.93, 1.23) | 0.370 | 1.07 (0.93, 1.23) | 0.359 |
| Q3 (N = 3102) | 1.87 (1.65, 2.13) | <0.001 | 1.77 (1.56, 2.01) | <0.001 | 1.34 (1.18, 1.52) | <0.001 |
| Q4 (N = 3102) | 3.52 (3.13, 3.96) | <0.001 | 3.21 (2.86, 3.61) | <0.001 | 1.84 (1.63, 2.08) | <0.001 |
| P for trend | | <0.001 | | <0.001 | | <0.001 |

Model 1 adjusted for none; Model 2 adjusted for age, sex, and race; Model 3 adjusted for age, sex, race, heart rate, respiratory rate, MBP, potassium, chloride, bicarbonate, anion gap, BUN, creatinine, glucose, HbA1c, CHF, atrial fibrillation, hypertension, cerebrovascular disease, COPD, liver disease, diabetes, renal disease, malignant tumor, medications used. BUN, blood urea nitrogen; CHF, congestive heart failure; COPD, chronic obstructive pulmonary disease.

Notably, significant interactions were detected for chronic pulmonary disease, norepinephrine use, and glucocorticoid use (all P for interaction < 0.05): the association was stronger in patients without chronic pulmonary disease (OR = 1.35, 95% CI: 1.26–1.45) than in those with chronic pulmonary disease (OR = 1.11, 95% CI: 1.00–1.24), more pronounced in patients not receiving norepinephrine (OR = 1.43, 95% CI: 1.31–1.55) compared to those receiving norepinephrine (OR = 1.12, 95% CI: 1.03–1.22), and similarly stronger in patients not receiving glucocorticoids (OR = 1.36, 95% CI: 1.26–1.48) than in glucocorticoid users (OR = 1.14, 95% CI: 1.04–1.25).

## Discussion

This study demonstrates that elevated lnPIV levels in critically ill patients with hyperlipidemia are significantly associated with all-cause mortality at 28, 90, and 180 days, and are also closely related to increased risks of acute kidney injury and delirium. This association remained highly robust in sensitivity analyses and across various clinical subgroups. Restricted cubic spline analysis revealed a nonlinear relationship between lnPIV and short-term mortality, with a marked risk increase at high lnPIV levels and minimal change at low to moderate levels, suggesting a potential threshold effect

**Table 3. Multivariable results by Logistic regression analysis.**

| Categories | Model1 | | Model2 | | Model3 | |
|---|---|---|---|---|---|---|
| | OR (95% CI) | P-value | OR (95% CI) | P-value | OR (95% CI) | P-value |
| In-hospital mortality | | | | | | |
| lnPIV | 1.44(1.39, 1.50) | <0.001 | 1.41(1.36, 1.47) | <0.001 | 1.26(1.19, 1.34) | <0.001 |
| Quartile | | | | | | |
| Q1 (N = 3102) | Reference | | Reference | | Reference | |
| Q2 (N = 3102) | 1.08 (0.88, 1.33) | 0.456 | 1.07 (0.87, 1.32) | 0.520 | 0.99 (0.79, 1.26) | 0.964 |
| Q3 (N = 3102) | 1.78 (1.47, 2.16) | <0.001 | 1.69 (1.40, 2.05) | <0.001 | 1.26 (1.02, 1.57) | 0.034 |
| Q4 (N = 3102) | 4.07 (3.43, 4.86) | <0.001 | 3.79 (3.19, 4.53) | <0.001 | 1.86 (1.51, 2.28) | <0.001 |
| P for trend | | <0.001 | | <0.001 | | <0.001 |
| ICU mortality | | | | | | |
| lnPIV | 1.46(1.39, 1.53) | <0.001 | 1.43(1.37, 1.50) | <0.001 | 1.20(1.12, 1.29) | <0.001 |
| Quartile | | | | | | |
| Q1 (N = 3102) | Reference | | Reference | | Reference | |
| Q2 (N = 3102) | 1.06 (0.82, 1.36) | 0.656 | 1.06 (0.82, 1.36) | 0.668 | 0.87 (0.65, 1.15) | 0.328 |
| Q3 (N = 3102) | 1.93 (1.54, 2.41) | <0.001 | 1.86 (1.49, 2.33) | <0.001 | 1.23 (0.95, 1.59) | 0.120 |
| Q4 (N = 3102) | 3.95 (3.22, 4.86) | <0.001 | 3.72 (3.03, 4.59) | <0.001 | 1.57 (1.23, 2.01) | <0.001 |
| P for trend | | <0.001 | | <0.001 | | <0.001 |
| AKI incidence | | | | | | |
| lnPIV | 1.13(1.10, 1.16) | <0.001 | 1.13(1.10, 1.16) | <0.001 | 1.10(1.05, 1.14) | <0.001 |
| Quartile | | | | | | |
| Q1 (N = 3102) | Reference | | Reference | | Reference | |
| Q2 (N = 3102) | 1.14 (1.01, 1.27) | 0.027 | 1.12 (1.00, 1.26) | 0.048 | 1.10 (0.97, 1.24) | 0.128 |
| Q3 (N = 3102) | 1.32 (1.18, 1.48) | <0.001 | 1.30 (1.16, 1.46) | <0.001 | 1.23 (1.08, 1.40) | 0.001 |
| Q4 (N = 3102) | 1.67 (1.48, 1.89) | <0.001 | 1.63 (1.45, 1.84) | <0.001 | 1.32 (1.16, 1.51) | <0.001 |
| P for trend | | <0.001 | | <0.001 | | <0.001 |
| Delirium incidence | | | | | | |
| lnPIV | 1.33(1.29, 1.37) | <0.001 | 1.32(1.28, 1.36) | <0.001 | 1.20(1.15, 1.25) | <0.001 |
| Quartile | | | | | | |
| Q1 (N = 3102) | Reference | | Reference | | Reference | |
| Q2 (N = 3102) | 1.13 (0.99, 1.28) | 0.067 | 1.14 (1.00, 1.29) | 0.051 | 1.12 (0.98, 1.28) | 0.086 |
| Q3 (N = 3102) | 1.88 (1.66, 2.12) | <0.001 | 1.87 (1.65, 2.11) | <0.001 | 1.38 (1.21, 1.57) | <0.001 |
| Q4 (N = 3102) | 2.97 (2.64, 3.34) | <0.001 | 2.92 (2.60, 3.29) | <0.001 | 1.79 (1.58, 2.04) | <0.001 |
| P for trend | | <0.001 | | <0.001 | | <0.001 |

Model 1 adjusted for none; Model 2 adjusted for age, sex, and race; Model 3 adjusted for age, sex, race, heart rate, respiratory rate, MBP, potassium, chloride, bicarbonate, anion gap, BUN, creatinine, glucose, HbA1c, CHF, atrial fibrillation, hypertension, cerebrovascular disease, COPD, liver disease, diabetes, renal disease, malignant tumor, medications used. BUN, blood urea nitrogen; CHF, congestive heart failure; COPD, chronic obstructive pulmonary disease.

at PIV ≈ 172.. Kaplan–Meier survival curves and cumulative hazard curves further demonstrated an increasing trend in mortality risk across lnPIV quartiles. ROC analysis indicated that the predictive performance of lnPIV was superior to that of several individual blood cell parameters (S2 Table). When compared with other composite inflammatory markers, PIV demonstrated competitive predictive performance, outperforming PLR and SII while showing comparable performance to NLR, MLR, and SIRI (S3 Table). Furthermore, adding PIV to the SOFA score significantly improved its discriminative ability. Decision curve analysis suggested that the SOFA+PIV model provided higher net benefit across clinically relevant

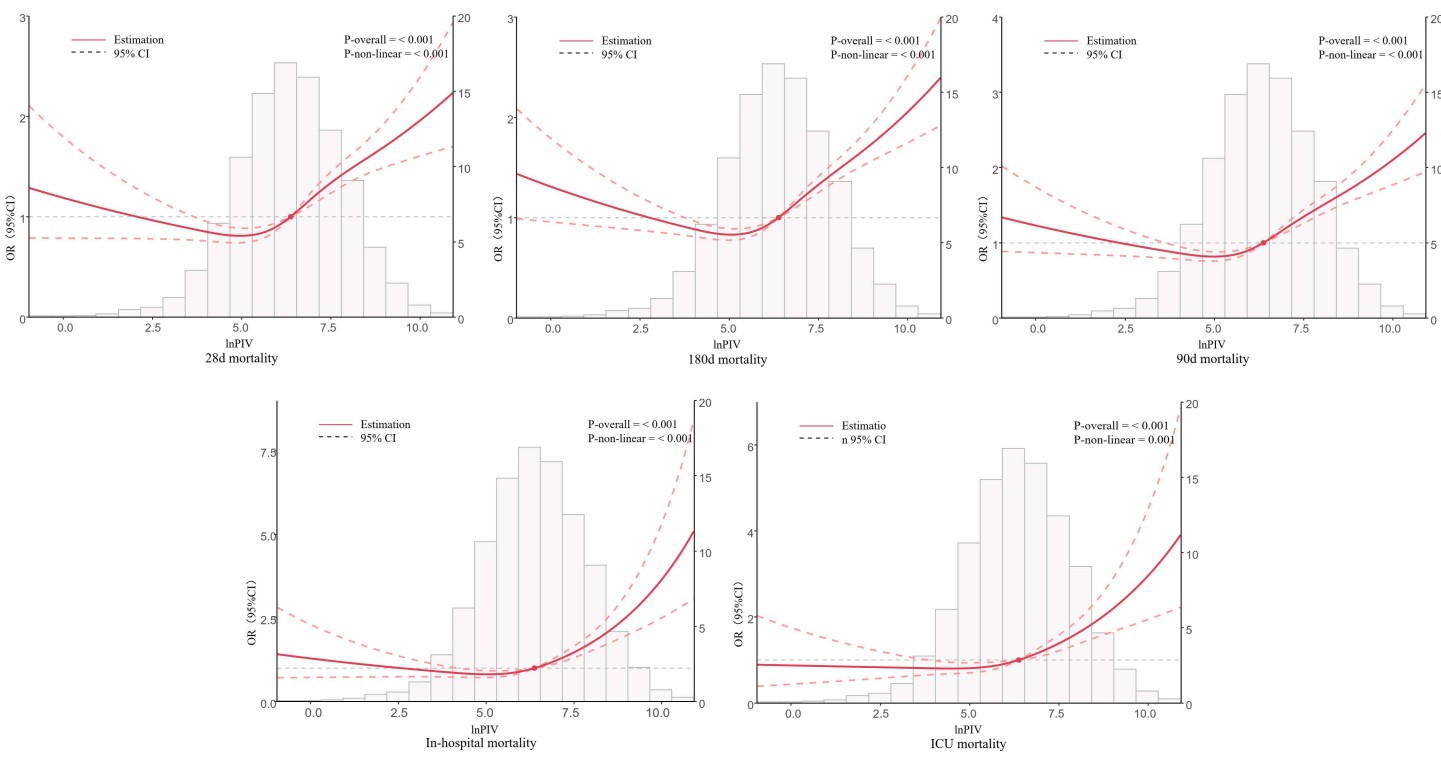

**Fig 3. Underlying non-linear correlations between lnPIV and all-cause mortality.**

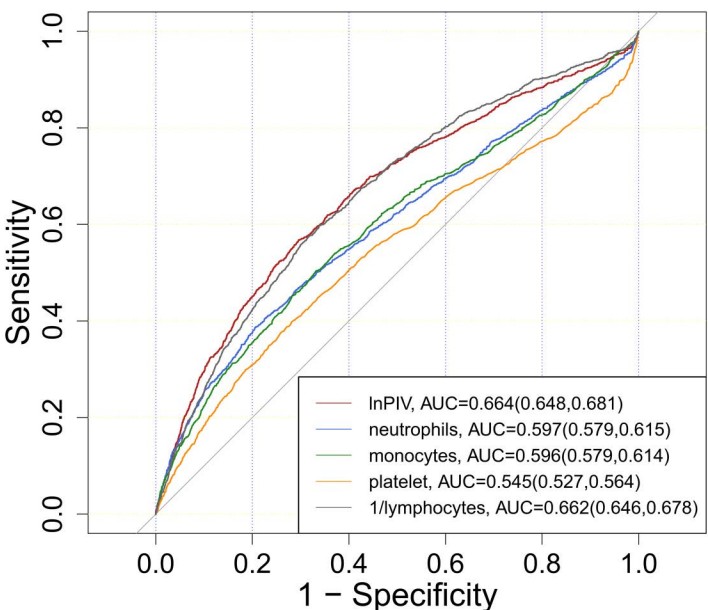

**Fig 4. ROC curve of the lnPIV to predict in-hospital mortality.**

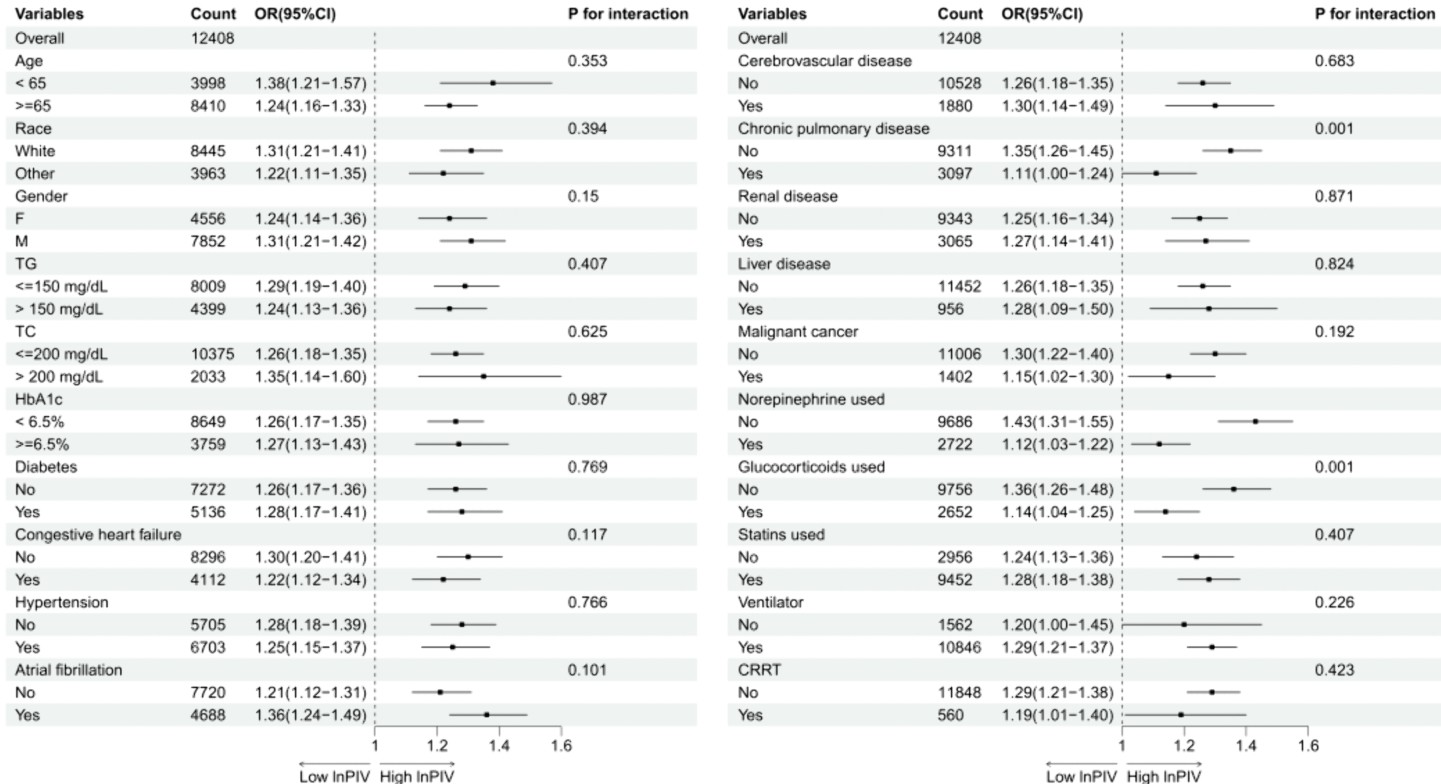

**Fig 5. Subgroup analysis for the effect of lnPIV on 28-day all-cause mortality.**

threshold probabilities compared to SOFA alone (S2 Fig). Subgroup analyses showed that the prognostic value of lnPIV was consistent across most clinical subgroups, but its effect was attenuated in patients with COPD and those receiving norepinephrine and glucocorticoids. Patients with COPD exhibit functional abnormalities and immune dysregulation in neutrophils, lymphocytes, and monocytes, which may obscure or interfere with the predictive ability of PIV as a composite inflammatory marker [21]. Glucocorticoids, often used for anti-inflammatory therapy, can also lead to alterations in blood cell counts, potentially influencing the PIV association [22].Patients requiring norepinephrine typically belong to a more severe subgroup with circulatory failure, where mortality risk is primarily driven by circulatory dysfunction, which may diminish the marginal predictive value of the immune-inflammatory burden captured by PIV [23]. Notably, existing literature has clearly demonstrated close associations between obesity (e.g., body mass index and waist circumference) and inflammation and dyslipidemia [24,25]. The relationship between obesity and critical illness remains controversial. Some studies suggest that obesity may exert a protective effect in certain specific diseases, but these conclusions require more rigorous interpretation [26,27]. The inability to include these obesity-related metrics due to data absence may compromise a comprehensive assessment of risk factors.

PIV encompasses a broader range of cell types compared to other inflammation-related indices (such as Platelet-to-Lymphocyte Ratio, Neutrophil-to-Lymphocyte Ratio, and Systemic Immune-Inflammation Index) [28], allowing for a more comprehensive reflection of the dynamic balance between inflammation and immunity. Consistent with our findings, all evaluated inflammatory markers significantly enhanced the predictive ability of the SOFA score (S4 Table). PIV consists of four blood cell types—neutrophils, lymphocytes, monocytes, and platelets—which directly participate in inflammatory responses, thrombosis, cytokine secretion, and immune regulation [29].

Neutrophils, as core cells of innate immunity, are major participants in the systemic inflammatory response [30,31]. Neutrophil counts increase markedly in sepsis, with excessive activation and delayed apoptosis, which can induce endo-thelial phenotypic changes and ultimately lead to microcirculatory dysfunction and organ failure [32,33].

Monocytes are a type of peripheral immune cell that participate in the inflammatory cascade by releasing cytokines and chemokines [34,35]. In patients with ischemic stroke, classical monocytes are closely associated with stroke severity and prognosis through disruption of the blood-brain barrier [36].

In addition to their traditional roles in hemostasis and thrombosis, platelets have significant immunoregulatory functions [37]. For example, via P-selectin, they bind neutrophils and monocytes to induce immune complex formation and amplify inflammation, a process demonstrated in infectious diseases such as HIV and dengue [38].

Lymphocytes are central to adaptive immunity and play important roles in both the initiation and resolution phases of inflammation, with their number and function regulated by various signaling pathways [39]. A decrease in lymphocyte count often suggests a poor prognosis; for example, sepsis can lead to lymphocytopenia [40–42]. However, as a single parameter, lymphocyte count captures only one aspect of the immune-inflammatory response.

This indicates that each cell type not only performs its individual functions, but also interacts with other cells to collectively reflect the host's inflammatory and immune status.

By integrating the roles of these four blood cell types, PIV can comprehensively reflect the host's inflammatory and immune status. This advantage has enabled PIV to expand from oncology and cardiovascular fields to studies of various critical illnesses. In studies of sepsis-associated acute kidney injury, elevated PIV independently predicts both short- and long-term mortality, and incorporating PIV into traditional scoring systems, such as SOFA or Acute Physiology And Chronic Health Evaluation II, can improve prognostic discrimination [43],consistent with our finding. Additionally, in studies of patients with aortic dissection, high PIV levels are significantly associated with an increase in postoperative complications and prolonged hospital stays [16]. Among patients with severe infections, such as sepsis and septic shock, the predictive ability of PIV surpasses that of commonly used single indicators, such as the neutrophil-to-lymphocyte ratio, further demonstrating its unique advantage in capturing complex immune-inflammatory pathways [14,44]. In young and overweight general populations, PIV shows a significant positive correlation with blood lipid levels, with reported values typically ranging from 300–600 in individuals with metabolic disorders [45]. However, it is important to emphasize that these findings are based on non-hospitalized populations. The substantially lower threshold observed in our cohort (PIV ≈ 172) highlights the profound impact of critical illness on the immune-inflammatory landscape, where even modest elevations carry significant prognostic implications. Further research is needed to validate the association between PIV and hyperlipidemia across different clinical settings. This study is the first to focus on this special population of critically ill patients with hyperlipidemia, showing that the overlap of metabolic and immune-inflammatory imbalance markedly amplifies risk and increases clinical management complexity. The data indicate that PIV outperforms conventional hematologic indices for early risk stratification, and its "threshold effect" with mortality suggests that dynamic monitoring may aid in identifying high-risk patients and facilitating targeted intervention. Furthermore, decision curve analysis demonstrated that combining PIV with SOFA provided greater clinical net benefit than SOFA alone, supporting its incremental value in clinical practice.

Although this study provides new insights into the prognostic role of PIV in managing critically ill patients with hyperlipidemia, several limitations remain. First, as a retrospective study, multivariable adjustments were performed, unmeasured or unknown confounders cannot be completely ruled out despite extensive multivariable adjustments. While we adjusted for a comprehensive set of demographic, clinical, laboratory, and medication variables, residual confounding from factors such as genetic predisposition, prior statin therapy duration, or detailed nutritional status may persist. Second, this was a single-center analysis based on data from a U.S. tertiary care hospital, which may limit generalizability to other settings. The predominantly White cohort (68.1%) and the specific healthcare system characteristics may not reflect populations with different racial/ethnic compositions, socioeconomic backgrounds, or healthcare access patterns. External validation

in multicenter, international, and more diverse populations is essential before widespread clinical application. Despite the large overall sample size, some subgroups (e.g., patients with hepatic or renal dysfunction) were underrepresented, which may have reduced statistical power. Third, although we performed multiple imputation to address missing lipid data and validated this approach with density plots (S1 Fig), the possibility of residual confounding from unmeasured lipid parameters (e.g., LDL-C, HDL-C, lipoprotein(a)) cannot be completely excluded. Additionally, the imputation process itself may introduce some uncertainty, and future studies with complete lipid profiles are warranted. Fourth, we assessed only the PIV level at ICU admission and did not include dynamic changes during hospitalization, limiting a comprehensive exploration of PIV trajectories. Fifth, while PIV demonstrated incremental value beyond SOFA (IDI = 0.04, NRI = 0.21), its modest individual discriminative ability (AUC ≈ 0.66) indicates that it should be used as a complementary tool alongside established clinical scores rather than as a standalone predictor. Finally, due to the observational design, this study cannot establish a causal relationship between elevated PIV levels and adverse outcomes, and prospective multicenter studies are needed to further elucidate its clinical mechanisms. Despite these limitations, the consistency of our findings across extensive sensitivity analyses, the biological plausibility of the associations, and the convergence with prior studies in other critically ill populations support the potential clinical utility of PIV as an adjunctive risk stratification tool in this vulnerable population.

## Conclusion

In critically ill patients with hyperlipidemia, elevated PIV is independently associated with increased risks of all-cause mortality, acute kidney injury, and delirium. Its predictive ability is superior to several traditional blood cell parameters, with a clinically relevant threshold identified at PIV ≈ 172.

While PIV demonstrates modest individual discriminative ability, it provides significant incremental value when combined with the SOFA score, improving risk classification and clinical net benefit. Patients with high PIV are more often older, female, and have multiple comorbidities, indicating greater clinical complexity. PIV may serve as a practical adjunctive tool for early risk stratification in this vulnerable population. Future multicenter prospective studies are needed to validate these findings and clarify the prognostic value of dynamic PIV changes.

## Supporting information

**S1 Table. Group differences by lnPIV quartiles.** Baseline characteristics of critically ill patients with hyperlipidemia stratified by lnPIV quartiles.
(DOCX)

**S2 Table. Predictive performance of PIV and individual blood cell parameters for in-hospital mortality.**
(DOCX)

**S3 Table. Predictive performance of PIV and other inflammatory markers for in-hospital mortality.**
(DOCX)

**S4 Table. Improvement in predictive performance by adding inflammatory markers to the SOFA score.**
(DOCX)

**S1 Fig. Density plots comparing observed and imputed values for key variables.** (A) Triglycerides, (B) Total cholesterol, (C) HDL, (D) HbA1c.
(TIF)

**S2 Fig. Decision curve analysis for in-hospital mortality.**
(TIF)

## Author contributions

**Data curation:** Xiao-Ying Huang.

**Formal analysis:** Xiao-Ying Huang.

**Investigation:** Xiao-Ying Huang.

**Methodology:** Xiao-Ying Huang.

**Software:** Xiao-Ying Huang.

**Visualization:** Xiao-Ying Huang.

**Writing – original draft:** Xiao-Ying Huang.

**Writing – review & editing:** Xiang Zheng.

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
