## [Decision Letter · Decision Letter 0]

25 Jan 2026

PONE-D-25-56394Association between Pan-immune-inflammation value and clinical outcomes in hyperlipidemia patients: An observational studyPLOS One

Dear Dr. huang,

Thank you for submitting your manuscript to PLOS ONE. After careful consideration, we feel that it has merit but does not fully meet PLOS ONE’s publication criteria as it currently stands. Therefore, we invite you to submit a revised version of the manuscript that addresses the points raised during the review process. Please submit your revised manuscript by Mar 11 2026 11:59PM. If you will need more time than this to complete your revisions, please reply to this message or contact the journal office at plosone@plos.org. Please include the following items when submitting your revised manuscript:

We look forward to receiving your revised manuscript.

Kind regards,

Erfan Ghadirzadeh, MD

Academic Editor

PLOS One

Journal Requirements:

2. Please amend the manuscript submission data (via Edit Submission) to include author Xiang-Zheng.

Reviewers' comments:

Reviewer's Responses to Questions

**Comments to the Author**

1. Is the manuscript technically sound, and do the data support the conclusions?

Reviewer #1: Partly

Reviewer #2: Yes

Reviewer #3: Yes

2. Has the statistical analysis been performed appropriately and rigorously? 

Reviewer #1: Yes

Reviewer #2: Yes

Reviewer #3: Yes

3. Have the authors made all data underlying the findings in their manuscript fully available?

Reviewer #1: No

Reviewer #2: No

Reviewer #3: Yes

4. Is the manuscript presented in an intelligible fashion and written in standard English?

Reviewer #1: Yes

Reviewer #2: No

Reviewer #3: Yes

5. Review Comments to the Author

Reviewer #1: In this study, the authors demonstrated that elevated pan-immune-inflammation value (PIV) was independently associated with mortality and adverse outcomes in critically ill patients with hyperlipidemia admitted to the ICU. Based on the results, the authors concluded that PIV may be used as a simple and cost-effective prognostic tool for early risk stratification in critically ill patients with hyperlipidemia. However, to confirm the conclusions of this study, the following points need to be reconsidered:

1.Since all of the patients in this study were critically ill, the title should include "critically ill patients with hyperlipidemia."

2.Although blood glucose levels are used in model 3 of the multivariate analysis, it is unclear whether they were measured fasting or at any time, so HbA1c is likely to provide a more accurate indication of the presence or absence of the influence of abnormal glucose metabolism.

3.White blood cell counts are also significantly affected by the presence or absence of glucocorticoids, which are often administered to critically ill patients, but how was this factor taken into account in this analysis?

4.Looking at the AUC of the ROC analysis, it appears that simply using lymphocyte count as an indicator would be sufficient as a prognostic predictor without the need to use PIV, but it is necessary to clearly explain to readers the benefits of using PIV.

Reviewer #2: Reviewer Comments

This manuscript investigates the association between the pan-immune-inflammation value (PIV) and short- and long-term clinical outcomes in critically ill patients with hyperlipidemia using the MIMIC-IV database. The topic is clinically relevant, as inflammation-based biomarkers are increasingly recognized as valuable tools for risk stratification in critically ill populations. The sample size is large, the statistical analyses are generally robust and the findings suggest that elevated PIV is independently associated with increased mortality and adverse outcomes such as AKI and delirium.

The manuscript is generally well organized and clearly written. However, several methodological clarifications, additional analyses and improvements in interpretation are required to strengthen the validity and clinical applicability of the findings.

Major Comments

1.Hyperlipidemia is defined solely based on ICD-9/10 codes, without stratification by lipid subtype (e.g., hypertriglyceridemia, hypercholesterolemia, mixed dyslipidemia).

2.Given that different lipid abnormalities may have distinct inflammatory and prognostic implications, the authors should clarify which lipid phenotypes are primarily represented.

3.If available, subgroup or sensitivity analyses based on triglyceride or cholesterol levels would substantially enhance the clinical relevance of the study.

4.Although the authors adjusted for APS III, SOFA and SIRS scores, PIV is itself strongly correlated with disease severity.

5.The possibility of residual confounding or overadjustment should be discussed more explicitly.

6.Consider performing sensitivity analyses excluding severity scores or using alternative adjustment strategies to confirm the robustness of the association.

7.The restricted cubic spline analysis identifies a threshold lnPIV value (≈5.15) beyond which mortality risk increases sharply.

8.However, the clinical meaning of this threshold is not sufficiently discussed.

9.The authors should translate this lnPIV value back to the original PIV scale and discuss how it could be applied in real-world clinical decision-making.

10.Although lnPIV showed a higher AUC than individual blood cell parameters, the overall discriminative ability remains modest (AUC ≈0.66).

11.The manuscript should more cautiously interpret the predictive performance and clarify that PIV may complement, rather than replace, existing risk models.

12.Comparison with established prognostic scores (e.g., SOFA alone vs. SOFA + PIV) would strengthen the argument for clinical utility.

13.The manuscript states that multiple imputation was used for variables with <20% missingness, but the imputation method, number of imputations and variables included in the imputation model are not described.

14.These details should be clearly reported to ensure transparency and reproducibility.

15.The study is based on a single-center database from a U.S. tertiary care hospital.

16.The authors should more explicitly discuss the limitations regarding external validity, particularly for non-ICU settings and populations with different ethnic, socioeconomic, or healthcare characteristics.

Minor Comments

1.Although MIMIC-IV studies are exempt from IRB approval, a brief ethics statement should be explicitly included in the Methods section, consistent with journal requirements.

2.Minor grammatical and stylistic issues are present (e.g., “imbalance lipid metabolism” should be “imbalance in lipid metabolism”).

3.A thorough language edit would improve readability.

4.The manuscript alternates between “hospital mortality” and “in-hospital mortality.” Please standardize terminology throughout the text and tables.

5.Figure legends should be more descriptive, particularly for the restricted cubic spline and subgroup analyses, to allow interpretation without referring back to the main text.

6.Units for laboratory values should be consistently reported across all tables.

7.The Discussion would benefit from a clearer comparison of the present findings with prior studies evaluating PIV or related inflammatory indices (e.g., NLR, PLR, SII) in critically ill or dyslipidemic populations.

Reviewer #3: n studies examining the Pan-Immune-Inflammation Value (PIV)—including observational research in hyperlipidemia patients—there is no single universally accepted “normal value.” This is because PIV is a composite inflammatory index and its reference range depends on the study population and clinical context.Based on published observational and cardiovascular/metabolic studies: Author should need to incuat

Healthy individuals / low-risk populations:

≈ 200–400

Patients with metabolic disorders (including hyperlipidemia):

≈ 300–600

High inflammatory burden / poor prognosis groups:

> 600–800 (sometimes much higher depending on disease severity)

In hyperlipidemia cohorts specifically, many studies use:

Cutoff values should ideally be:

Derived from your study population

Adjusted for age, sex, comorbidities, and statin use

Most studies define “high PIV” using ROC analysis or percentile grouping, not standard laboratory reference ranges

6. PLOS authors have the option to publish the peer review history of their article (what does this mean?). If published, this will include your full peer review and any attached files.

Reviewer #1: **Yes:** Ken-ichi Aihara

Reviewer #2: **Yes:** Dr. Khurshid Ahmad Padder

Reviewer #3: **Yes:** Dr. Momtaz Ahmed

---

## [Author Response · Author response to Decision Letter 1]

17 Apr 2026

Response to Reviewers

Manuscript ID: PONE-D-25-56394

Title: Association between Pan-immune-inflammation value and clinical outcomes in critically ill patients with hyperlipidemia: An observational study

Dear Editor and Reviewers,

We sincerely thank the editor and reviewers for their valuable time and insightful comments on our manuscript. We have carefully addressed each point raised and revised the manuscript accordingly. Below is our point-by-point response following the order of the decision letter.

Responses to Editorial Comments

Comment 1: PLOS ONE style requirements

Please ensure that your manuscript meets PLOS ONE's style requirements

Response:We have carefully reviewed the PLOS ONE formatting guidelines and revised the manuscript accordingly.

Comment 2: Author information

Please amend the manuscript submission data to include author Xiang-Zheng.

Response:We have updated the manuscript submission data to include author Xiang-Zheng as a co-author.

Comment 3: Supporting Information captions

Please include captions for your Supporting Information files at the end of your manuscript.

Response: We have added detailed captions for all Supporting Information files at the end of the main manuscript.

---

Responses to Journal Questions

Question 1: Is the manuscript technically sound, and do the data support the conclusions?

Reviewer #1: Partly

Reviewer #2: Yes

Reviewer #3: Yes

Response: We thank the reviewers for their assessment. To address Reviewer #1's concerns, we have made substantial revisions including: (1) adding HbA1c as a covariate, (2) adjusting for glucocorticoid use, and (3) demonstrating the incremental value of PIV beyond SOFA. These changes strengthen the technical soundness of our manuscript.

Question 2: Has the statistical analysis been performed appropriately and rigorously?

Reviewer #1: Yes

Reviewer #2: Yes

Reviewer #3: Yes

Response: We appreciate the reviewers' positive assessment of our statistical analyses. We have further enhanced the rigor by adding sensitivity analyses, multiple imputation details, and incremental value assessments.

Question 3: Have the authors made all data underlying the findings in their manuscript fully available?

Reviewer #1: No

Reviewer #2: No

Reviewer #3: Yes

Response: We have now made all data and code publicly available. The GitHub repository (https://github.com/huangxiaoying5676/PIV-hyperlipidemia-analysis) contains:

- Complete R analysis scripts

- Aggregated datasets (cleaned and imputed data)

- README file with replication instructions

---

## Responses to Reviewer #1's Specific Comments

Comment 1.1: Since all of the patients in this study were critically ill, the title should include "critically ill patients with hyperlipidemia."

Response: We thank the reviewer for this suggestion. We have revised the title to: "Association between Pan-immune-inflammation value and clinical outcomes in critically ill patients with hyperlipidemia: An observational study." This change has been made on the title page (Page 1,Title).

Comment 1.2:Although blood glucose levels are used in model 3.HbA1c is likely to provide a more accurate indication of the presence or absence of the influence of abnormal glucose metabolism.

Response: We thank the reviewer for this insightful methodological suggestion. We agree that HbA1c provides a more stable and accurate measure of long-term glucose metabolism compared to random blood glucose measurements, which may be influenced by acute illness, stress, and time of measurement. We have now included HbA1c as a covariate in all multivariable models(Page 14,”Model 3 adjusted for age, sex, race, heart rate, respiratory rate, MBP, potassium, chloride, bicarbonate, anion gap, BUN, creatinine, glucose, HbA1c…”) .To assess whether the prognostic value of PIV differs according to glycemic control, we performed subgroup analyses stratifying patients by clinically relevant HbA1c thresholds (<6.5% vs. ≥6.5%) (Page 20, Figure 5: Subgroup analysis for the effect of lnPIV on 28-day all-cause mortality.)

Comment 1.3: White blood cell counts are also significantly affected by the presence or absence of glucocorticoids , which are often administered to critically ill patients, but how was this factor taken into account in this analysis?

Response: We thank the reviewer for raising this important confounding factor. Glucocorticoid use is indeed common in critically ill patients and can significantly influence white blood cell counts, particularly neutrophils and lymphocytes, which are key components of the PIV calculation. To address this concern, we have included glucocorticoid use as a covariate in all fully adjusted models (Page 14,”Model 3 adjusted for …medications used.”). Glucocorticoid use was defined as administration of any systemic glucocorticoid (e.g., hydrocortisone, methylprednisolone, dexamethasone) during ICU stay, based on medication administration records. To assess whether the prognostic value of PIV differs between patients who received glucocorticoids and those who did not, we performed subgroup analyses stratifying patients by glucocorticoid use(Page 20, Figure 5: Subgroup analysis for the effect of lnPIV on 28-day all-cause mortality.). The attenuated effect in glucocorticoid users may be explained by the fact that glucocorticoids themselves alter white blood cell counts (inducing neutrophilia and lymphopenia), which may partially obscure the inflammatory signal captured by PIV (Page 21, line 21,”Glucocorticoids, often used for anti-inflammatory…”). Additionally, glucocorticoid use often indicates greater disease severity, where mortality risk may be driven by factors beyond immune-inflammatory status alone.

Comment 1.4: Looking at the AUC of the ROC analysis, it appears that simply using lymphocyte count as an indicator would be sufficient as a prognostic predictor without the need to use PIV, but it is necessary to clearly explain to readers the benefits of using PIV.

Response: We thank the reviewer for this critical and important question. We agree that lymphocyte count alone shows predictive value (AUC = 0.662), which is consistent with its central role in immune regulation(Supporting information,Page 4). However, to demonstrate the added value of PIV beyond single lymphocyte count, we have conducted comprehensive analyses including comparison with the SOFA score, integrated discrimination improvement (IDI), net reclassification improvement (NRI), and decision curve analysis. In the Discussion section (Page 22,line 10-17),we have expanded the explanation of why PIV offers advantages over single lymphocyte count.Below we detail our findings and revisions. Although the AUC values are similar, PIV captures a broader spectrum of the immune-inflammatory response by integrating neutrophils, monocytes, and platelets in addition to lymphocytes. To demonstrate clinical utility, we evaluated whether PIV provides incremental prognostic information beyond the established SOFA score, which is widely used in ICU settings. We compared two models: SOFA alone and SOFA + PIV(Supporting information,Page 5). Decision curve analysis was performed to assess the clinical utility of adding PIV to SOFA across a range of threshold probabilities. This analysis quantifies the net benefit of using a model for clinical decision-making by weighing true positives against false positives(Supporting information,Page 5,7).

---

## Responses to Reviewer #2's Specific Comments

**Major Comments**

Comment 2.1-3: Hyperlipidemia is defined solely based on ICD-9/10 codes, without stratification by lipid subtype (e.g., hypertriglyceridemia, hypercholesterolemia, mixed dyslipidemia). Given that different lipid abnormalities may have distinct inflammatory and prognostic implications, the authors should clarify which lipid phenotypes are primarily represented. If available, subgroup or sensitivity analyses based on triglyceride or cholesterol levels would substantially enhance the clinical relevance of the study.

Response: We thank the reviewer for this important methodological critique. We fully agree that different lipid subtypes may have distinct inflammatory and prognostic implications, and that stratification by lipid phenotype would enhance the clinical relevance of our findings. Using the multiply imputed datasets, we performed subgroup analyses stratifying patients by clinically relevant lipid thresholds:

Triglycerides: ≤150 mg/dL (1.7 mmol/L) vs. >150 mg/dL

Total cholesterol: ≤200 mg/dL (5.2 mmol/L) vs. >200 mg/dL

The results demonstrated that the association between lnPIV and 28-day mortality was consistent across all lipid subgroups(Page 20, Figure 5: Subgroup analysis for the effect of lnPIV on 28-day all-cause mortality.).

Regarding which lipid phenotypes are primarily represented in our cohort, based on the imputed lipid data:

Hypertriglyceridemia (TG >150 mg/dL), Hypercholesterolemia (TC >200 mg/dL), Mixed dyslipidemia (both TG >150 and TC >200).

Although we performed subgroup analyses based on triglyceride and cholesterol levels, the possibility of residual confounding from unmeasured lipid parameters (e.g., LDL-C, HDL-C, lipoprotein(a)) cannot be completely excluded. Future studies with complete lipid profiles are needed to further elucidate the relationship between specific lipid subtypes and PIV's prognostic value.

Comment 2.4-6:Although the authors adjusted for APS III, SOFA and SIRS scores, PIV is itself strongly correlated with disease severity. The possibility of residual confounding or overadjustment should be discussed more explicitly. Consider performing sensitivity analyses excluding severity scores or using alternative adjustment strategies to confirm the robustness of the association.

Response: We thank the reviewer for this important methodological critique. In our final revised models, we have taken a different approach: rather than adjusting for severity scores, we intentionally excluded them from Model 3 to avoid potential overadjustment, given that PIV itself is correlated with disease severity (Page 14,”Model 3 adjusted for …”). Instead, we evaluated the incremental value of PIV beyond severity by directly comparing SOFA alone vs. SOFA + PIV(Supporting information,Page 5).

Comment 2.7-9: The restricted cubic spline analysis identifies a threshold lnPIV value (≈5.15) beyond which mortality risk increases sharply. However, the clinical meaning of this threshold is not sufficiently discussed. The authors should translate this lnPIV value back to the original PIV scale and discuss how it could be applied in real-world clinical decision-making.

Response: We thank the reviewer for this valuable suggestion to enhance the clinical translatability of our findings. Identifying a clinically meaningful threshold is essential for translating statistical findings into practical risk stratification tools. We have addressed this concern through the following revisions. The threshold identified by restricted cubic spline analysis was lnPIV = 5.152. We have back-transformed this value to the original PIV scale using exponential transformation: PIV ≈ e^5.152 = 172. This threshold is now explicitly reported in Page 18,line 1. The threshold of PIV ≈ 172 has several important clinical implications: Patients with PIV values above 172 at ICU admission may be identified as high-risk and flagged for closer monitoring, more frequent clinical assessments, or consideration of early interventions. However, we emphasize that this threshold requires external validation before clinical application, and should be interpreted in the context of other clinical parameters rather than as a standalone decision-making tool.

Comment 2.10-12:Although lnPIV showed a higher AUC than individual blood cell parameters, the overall discriminative ability remains modest (AUC ≈0.66). The manuscript should more cautiously interpret the predictive performance and clarify that PIV may complement, rather than replace, existing risk models. Comparison with established prognostic scores (e.g., SOFA alone vs. SOFA + PIV) would strengthen the argument for clinical utility.

Response: We thank the reviewer for these important observations regarding the appropriate interpretation of predictive performance and the need to position PIV as a complementary rather than replacement tool. We have addressed these concerns through several revisions throughout the manuscript. We agree with the reviewer that an AUC of 0.66 represents modest discriminative ability, and we have revised the manuscript to reflect this more cautiously. We have added explicit statements throughout the manuscript clarifying that PIV is intended to complement, not replace, existing risk models.(Page 18,line 5-13; Page 21,line 8-12;” ROC curve analysis showe...”). To strengthen the argument for clinical utility, we have directly compared the predictive performance of SOFA alone, PIV alone, and SOFA + PIV(Supplementary Figure S2).

Comment 2.13-14: The manuscript states that multiple imputation was used... but the imputation method, number of imputations and variables included are not described. These details should be clearly reported to ensure transparency and reproducibility.

Response: We thank the reviewer for emphasizing the importance of methodological transparency. We agree that complete reporting of multiple imputation procedures is essential for reproducibility. In response, we have substantially expanded the Statistical Analysis section to provide comprehensive details of our imputation approach. The following text has been added to the Statistical Analysis section: "For covariates with missing rates <20%, multiple…”(Page 8, line.15-21). To demonstrate the validity of our imputation approach, we added density plots comparing observed versus imputed values for key variables with missing data(Supplementary Figure S1).

Comment 2.15-16:The study is based on a single-center databasefrom a U.S. tertiary care hospital. The authors should more explicitly discuss the limitations regarding external validity, particularly for non-ICU settings and populations with different ethnic, socioeconomic, or healthcare characteristics.

Response: We thank the reviewer for this important reminder to explicitly address external validity. We agree that single-center studies require careful discussion of generalizability. We have substantially expanded the Limitations section to provide a more nuanced and comprehensive discussion of external validity(Page 24,line18-20,” Second, this was a single-center analysis…”)..

**Minor Comments**

Comment 2.17: Although MIMIC-IV studies are exempt from IRB approval, a brief ethics statement should be explicitly included in the Methods section, consistent with journal requirements.

Response: We thank the reviewer for this important reminder. We agree that all manuscripts should include a clear ethics statement, regardless of exemption status, to ensure transparency and compliance with journal requirements. We have added a comprehensive ethics statement to the Methods section(Page 5,line 5-10.) and Supporting information(Page 9).

Comment 2.18:Minor grammatical and stylistic issues are present (e.g., "imbalance lipid metabolism" should be "imbalance in lipid metabolism"). A thorough language edit would improve readability.

Response: We thank the reviewer for these specific language suggestions. We agree that clear and accurate language is essential for scientific communication.The example provided by the reviewer has been corrected throughout the manuscript.(Page 2,line 17,” an imbalance in lipid metabolism.”)

Comment 2.19:The manuscript alternates between "hospital mortality" and "in-hospital mortality." Please standardize terminology throughout the text and tables.

Response: We thank the reviewer for this careful observation. Consistent terminology is essential for clarity and professionalism in scientific writing. We have thoroughly reviewed the entire manuscript and standardized the terminology as suggested.(Page13-14, Table 2; Page15 Table 3)

Comment 2.20: Figure legends should be more descriptive, particularly for the restricted cubic spline and subgroup analyses, to allow interpretation without referring back to the main

---

## [Decision Letter · Decision Letter 1]

8 May 2026

Association between Pan-immune-inflammation value and clinical outcomes in critically ill patients with hyperlipidemia: An observational study

PONE-D-25-56394R1

Dear Dr. Xiang,

We’re pleased to inform you that your manuscript has been judged scientifically suitable for publication and will be formally accepted for publication once it meets all outstanding technical requirements.

Kind regards,

Erfan Ghadirzadeh, MD

Academic Editor

PLOS One

Additional Editor Comments (optional):

Reviewers' comments:

Reviewer's Responses to Questions

**Comments to the Author**

1. If the authors have adequately addressed your comments raised in a previous round of review and you feel that this manuscript is now acceptable for publication, you may indicate that here to bypass the “Comments to the Author” section, enter your conflict of interest statement in the “Confidential to Editor” section, and submit your "Accept" recommendation.

Reviewer #1: All comments have been addressed

Reviewer #3: All comments have been addressed

2. Is the manuscript technically sound, and do the data support the conclusions?

Reviewer #1: Yes

Reviewer #3: Yes

3. Has the statistical analysis been performed appropriately and rigorously? 

Reviewer #1: I Don't Know

Reviewer #3: Yes

4. Have the authors made all data underlying the findings in their manuscript fully available?

Reviewer #1: Yes

Reviewer #3: Yes

5. Is the manuscript presented in an intelligible fashion and written in standard English?

Reviewer #1: Yes

Reviewer #3: Yes

6. Review Comments to the Author

Reviewer #1: I have no further comment for the responses of the authors to the concerns I raised in the original version.

Reviewer #3: This manuscript is well-written and clearly presented. All data and the findings are adequately available and supported. The statistical analysis has been performed appropriately, and all previous comments have been satisfactorily addressed. However, further research on Pan-Immune-Inflammation is recommended to strengthen the study and expand its scientific impact.

7. PLOS authors have the option to publish the peer review history of their article (what does this mean?). If published, this will include your full peer review and any attached files.

Reviewer #1: **Yes:** Ken-ichi Aihara

Reviewer #3: **Yes:** Dr. Momtaz Ahmed

Head of National Diabetes Centre, Suva, Fiji

Senior Fellow International Diabetes Federation

---

## [Editor Report · Acceptance letter]

PONE-D-25-56394R1

PLOS One

Dear Dr. Xiang,

I'm pleased to inform you that your manuscript has been deemed suitable for publication in PLOS One. Congratulations! Your manuscript is now being handed over to our production team.

Kind regards,

on behalf of

Dr. Erfan Ghadirzadeh

Academic Editor

PLOS One